# The Implication of the First Wave of COVID-19 on Mental Health: Results from a Portuguese Sample

**DOI:** 10.3390/ijerph19116489

**Published:** 2022-05-26

**Authors:** Jorge Quintas, Ana Guerreiro, Maria João Leote de Carvalho, Vera Duarte, Ana Rita Pedro, Ana Filipa Gama, Inês Keygnaert, Sónia Dias

**Affiliations:** 1CJS—Interdisciplinary Research Centre on Crime, Justice and Security, School of Criminology, Faculty of Law, University of Porto, 4050-123 Porto, Portugal; ana.esteves.guerreiro@gmail.com; 2Department of Social and Behavioral Sciences, University of Maia (UMAIA), 4475-690 Maia, Portugal; vduarte@ismai.pt; 3NOVA School of Social Sciences and Humanities (NOVA FCSH), 1069-061 Lisbon, Portugal; mjleotec@sapo.pt; 4CICS.NOVA—Interdisciplinary Centre of Social Sciences, 1099-085 Lisbon, Portugal; 5Public Health Research Centre, NOVA National School of Public Health, Universidade NOVA de Lisboa, 1600-560 Lisbon, Portugal; rita.pedro@ensp.unl.pt (A.R.P.); anafgama@gmail.com (A.F.G.); sonia.dias@ensp.unl.pt (S.D.); 6Comprehensive Health Research Centre (CHRC), Universidade NOVA de Lisboa, 1150-082 Lisbon, Portugal; 7Department of Public Health and Primary Care, Faculty of Medicine and Health Sciences, Ghent University, 9000 Ghent, Belgium; ines.keygnaert@ugent.be; 8WHO Collaborating Centre: International Centre for Reproductive Health (ICRH), Centre for Social Studies on Migration and Refugees (CESSMIR), 9000 Ghent, Belgium

**Keywords:** COVID-19, lockdown, mental health, public health, psychological impact

## Abstract

The social conditions created by the COVID-19 pandemic had a great potential to affect the mental health of individuals. Meta-analyses indicate a rise in these problems in these periods among general populations, patients and health professionals, even with substantial heterogeneous results. This paper examines mental health impacts specifically during the first wave of COVID-19. An online survey was conducted with a Portuguese convenience sample (N = 1.062) comprising questions about substance use, perceived stress, post-traumatic stress disorder and self-damage behaviors. The results concerning substance use show an extensive use of medication to sleep or calm down, especially among women and older respondents, a small percentage of alcohol consumers with a high pattern of use and less frequent cannabis consumption, even with a quarter of users who began only in the COVID-19 period. The rates of perceived stress and PTSD were higher compared with international prevalence estimations during the pandemic conditions. Both correlated measures were worse for women and young people. Another problematic issue was the rate of suicidal ideation, with a relevant proportion of starters during this period. These data reinforce the need to promote access to mental health services.

## 1. Introduction

Worldwide, in the last two years, the COVID-19 pandemic has severely affected societies and most people’s lives, posing a wide range of mental health risks [1,2]. In January 2020 the World Health Organization (WHO) declared the outbreak of a new coronavirus disease caused by the virus SARS-CoV-2 as a public health emergency of international concern. Immediately, several states initiated coordinated actions aiming to prevent and combat this new disease, although social, economic, political and educational consequences, among others, could ensue [3,4,5]. On 11 March 2020, due to its increasingly global dissemination, the WHO declared COVID-19 to be a pandemic [6]. In Portugal, the first COVID-19 case was confirmed on 2 March 2020. Two weeks later, on 19 March 2020, the Portuguese Government imposed the first state of emergency, enforcing a set of restrictive measures until 2 May 2020, including quarantine and social distancing [7]. After this period and during the summer months, the restrictions were reduced until the beginning of a second wave of infections, bringing a new declaration of a state of emergency on 6 November 2020.

The potential impact of this public health emergency on the mental health of general society was soon recognized by health authorities, given the fear raised by the rapid spread of the virus and the high figures on related deaths as well in face of the significant constraints on social interaction imposed by the prophylactic measures taken to control the outbreak [8]. The ways individuals live and socialize, including basic daily routines, study and work, suddenly deeply changed, influencing emotional and psychosocial well-being [9,10].

The call for urgent actions led the WHO Department of Mental Health and Substance to advance on 18 March 2020 with a statement on mental health and psychosocial considerations to support the well-being of specific target groups and the general population during the COVID-19 outbreak [11].

The expected lasting effects of this pandemic on mental health have raised particular concern. COVID-19 patients, especially those with severe illness conditions, had to deal with serious concerns about the progression of the disease or even directly suffer several psychiatric symptoms. Healthcare professionals faced relevant stressors at the workplace (e.g., continuous daily workload and risks of their own health or even life). Worry about personal, professional and financial situations have crossed most people’s thoughts. Grief due to personal losses is present in millions of families worldwide. Moreover, the general population have been affected by the restrictive measures (e.g., quarantine or lockdown) or, at least, the public health measures (e.g., outdoor masks, physical distance and vaccines) that increase social isolation, economic deprivations, fears of contamination and insecurity [12,13,14]. The emotional fallout of social distancing is intense and diverse. Risk factors associated with potential suicidal behaviors (e.g., trauma, abuse, social isolation, job and financial losses and mental health disorders, among others) in association with barriers to accessing basic services, including health, have been exacerbated during this pandemic [15,16,17,18,19].

Previous research on the impact of quarantine measures enforced in different world regions due to the spread of other pandemics in the last decades—SARS-CoV-1 (severe acute respiratory syndrome), MERS-Cov (Middle East respiratory syndrome), Ebola and H1N1 (influenza A virus subtype)—identified a wide range of mental health problems affecting individuals [7,20,21]. A variety of psychological symptoms, such as stress, anxiety, fear, depression or even post-traumatic stress disorder (PTSD), were identified as common throughout a pandemic time [12,22]. For instance, PTSD [23,24] emerged because of the disease but was also associated with the level of uncertainty in people’s lives and was experienced as result of intense mediatization and news on social media, which prolonged periods of stress [25,26].

Recent meta-analyses, e.g., [27,28,29,30], also show how COVID-19 pandemic conditions affected the mental health of COVID-19 patients, healthcare professionals and the general population. A study [28] examining several short-term mental health consequences of COVID-19 (K = 55; N = 189.159) found prevalences of depression (k = 46) 15.97% (95%CI, 13.24–19.13), anxiety (k = 54) 15.15% (95%CI, 12.29–18.54), insomnia (k = 14) 23.87% (95%CI, 15.74–34.48), PTSD (k = 13) 21.94% (95%CI, 9.37–43.31) and psychological distress (k = 19)13.29% (95%CI, 8.80–19.57) that were significantly higher compared to the general population under normal circumstances. Specifically, the study found that the prevalence of depression symptoms in populations affected by COVID-19 was more than three times higher (15.97%) compared to the general population (4.4%), four times higher for anxiety (15.15% vs. 3.6%) and five times higher for PTSD (21.94% vs. 4%). These consequences were equally high across populations and affected countries and across gender. The only reports of insomnia are significantly higher among health care professionals than the general population.

Focusing on PTSD symptoms, a retrospective study including 3405 participants estimated 17.68% to be the overall prevalence of PTSD in COVID-19 patients [27]. In severe cases of COVID-19, the prevalence of PTSD was slightly higher at 19% (K = 2; N = 200 patients). However, longer-term COVID-19 patients typically exhibited no worse than mild symptoms of anxiety and depression, and the prevalence of these symptoms were consistent with the general population. Moreover, another study examining only survivors of severe COVID-19 infection concluded that a PTSD pooled prevalence of 16% (9 to 25)-k = 13; N = 1093) seems to exist [31]. Research with a larger meta-analysis (k = 63; from 24 counties; N = 124.952) found an overall pooled estimate of PTSD prevalence of 17.52% (13.89–21.86) [30]. The authors [30] also found that, interestingly, the population at large (17.34%) and health professionals (17.23%) have a higher estimation of PTSD prevalence when compared with patients (15.45%). This result was not found in another meta-analysis developed with 88 studies about healthcare professionals [29]. The authors concluded that healthcare workers had the highest prevalence of PTSD (26.9%; 95% CI: 20.3–33.6%), followed by infected cases (23.8%: 16.6–31.0%) and the general public (19.3%: 15.3–23.2%). Other research [30] also concluded that some specific health care professionals have higher PTSD estimations, namely, covid unit workers (30.96%) and nurses (28.22%). Other interesting data showed that, contrary to expectations, elderly individuals present a lower prevalence compared with other adults.

The long-term consequences of the lockdown on mental health are not fully understood, but a body of research suggests that being in lockdown is associated with poor social and emotional well-being, especially in children, e.g., [32,33]. For example, a study analyzing the impact of the COVID-19 lockdown on children’s mental health in Spain and Italy found that more than 85% of the parents reported changes in their children’s emotional state, particularly concentration difficulties (76.6%) [34]. These changes can also be a result of the parent’s perceived stress by children [35]. Research about the psychological effects of the COVID-19 lockdown on children and their families in the UK [36] showed a high percentage of significant changes in children’s emotional states and behavior during the lockdown. The children mostly felt bored (73.8%), lonely (64.5%), frustrated (61.4%), irritable (57.1%), worried (52.4%) and sad (43.4%) [36].

The current pandemic crisis has quickly become global on a scale hardly seen before, with the need to update understanding and knowledge on its evolution becoming key as new challenges and issues promptly emerge [37]. There has been a massive and prolonged enforcement of quarantine measures worldwide in reaction to successive waves and peaks in virus dissemination [38]. This situation has challenged individual resilience and mental health as well as social cohesion. There is a growing body of knowledge that states that the longer people are isolated, the greater the risk of suffering psychiatric diseases, namely, depression, irritability, anxiety, fear, anger and insomnia, e.g., [13,32,39,40,41].

For most of the population, there was an intense daily coexistence with all family members who lived in the same house as had never happened before [36]. However, this presence in a(n) (apparent) safe environment also had other consequences, namely, on familiar pressure that could lead to mental health problems. Intergenerational relations were strongly affected, as most of the elderly people, considered to be a highly vulnerable group, became deprived of personal contacts and attention from their own families for a long time. Moreover, the attempt to conciliate the personal, familial and professional dimensions of one individual’s life was affected by the external factors and stressors that caused internal changes, which may have contributed to increased fear and anxiety [4,24]. Schools moved to online digital platforms whenever possible, which meant many children in different parts of the world were excluded since not all had access to technological means. To both children and elderly placed in institutional residential care, social distancing measures were aggravated, subjecting individuals to increased social isolation. Many services and work tasks moved online, but others shut down, and unemployment has grown significantly. On the other hand, essential workers have never stopped and kept on going in their daily routines at the frontline of the pandemic.

Not all individuals react and manage similar mental health risk factors and stressors in the same way. Each person is influenced by specific factors, from individual characteristics (personal resources, personality and previous experiences) to external factors, such as the social and environmental contexts, community and familial support, social networks and policy practices, among others [24,42]. It is the interaction of these influences that shapes the way one goes through and overcomes adverse experiences. The fears associated with the pandemic, first of one becoming infected and dying or losing a relative, and second, of one being kept under confinement measures for an indefinite period of time without the possibility of living his/her own live freely, can be the triggers of an individual’s mental health problems [28,43].

The Portuguese situation deserves special attention, given the high prevalence of mental health problems in the general population before the pandemic outbreak. According to official data from the Health National Council [44] and from the Portuguese Society of Psychiatry and Mental Health [45], in Portugal (i) psychiatric disorders have a prevalence of 22.9%, placing the country in a worrying second place among the European countries; (ii) mental and behavioral disorders represent 11.8% of the global burden of disease in Portugal, more than oncological diseases (10.4%) and only surpassed by cerebrovascular diseases (13.7%); (iii) among psychiatric disorders, anxiety disorders have the highest prevalence (16.5%), and depression affected 10% of the Portuguese population, as impulse control disorders and substance abuse disorders have lower prevalence rates, respectively, with 3.5% and 1.6% prevalences; (iv) additionally, dementia has a frequency of 20.8 per 1000 inhabitants, which places Portugal in 4th place among the Organisation for Economic Co-operation and Development (OECD) countries; and (v) as for the consumption of antidepressant drugs, Portugal places 5th in the OECD, also showing significant figures in relation to the consumption of anxiolytics. Moreover, in 2019, the suicide rate per 100.000 inhabitants was 9.7, which was almost four times higher in men (15.5) than in women (4.4). The figures concerning the suicide rate among the elderly population over 65 years old are above the national average in both men (32.6) and women (8.1) [46].

In the face of the lack of evidence on the mental health impact of this “new” disease, the NOVA National School of Public Health, in collaboration with Ghent University, launched a research project about violence and the impact of COVID-19 [16], which was developed with other Portuguese university partners. The project was focused on intimate partner violence and aimed to better understand the impact of the lockdown measures on the Portuguese population, shedding light on the self-reported occurrence of domestic violence and associated sociodemographic factors as well as on mental health problems related to pandemic times. Drawing from data from this wider project, the present study intends to explore the mental health impact of COVID-19 disease in the Portuguese population.

## 2. Materials and Methods

The data were collected using an online survey with closed-ended questions that was conducted between April and October 2020, administered through the Qualtrics software program (Qualtrics, Provo, UT, USA). The survey was disseminated by partner networks, digital social networks, social media and community institutions. After clicking on the link for the survey, potential participants accessed information on the study and the consent form. Only respondents who gave informed consent were able to participate in the survey.

For this study, the inclusion criteria were being >16 years old and living in Portugal. The questionnaire was developed by Ghent University based on the UN-MENAMAIS Study questionnaire [47] and drew from a set of validated instruments [48,49,50,51], including the evaluation of post-traumatic stress disorder (PC-PTSD-5) [52] and the Perceived Stress Scale [53]. Participants were asked about their sociodemographic characteristics and mental health problems, namely, substance use, perceived stress, post-traumatic stress disorder and self-damage questions. All mental health questions were asked concerning the period of the previous four weeks. Domestic violence questions (not analyzed in this article) included the experience during the COVID-19 pandemic. Specifically, participants were asked whether situations of physical, sexual and psychological violence had occurred. Minor adaptations were made to the national context in terms of language and the response options to some questions (e.g., level of education). To ensure cross-country comparability, all proposed changes were accepted by both countries’ research teams. The questionnaire was pretested among a convenience sample of individuals who met the inclusion criteria to ensure comprehensibility and to solve operational errors.

A descriptive analysis was performed using IBM-SPSS Statistics version 26 to analyze the sociodemographic characteristics and mental health outcomes. A bivariate analysis between sociodemographics and mental health, including substance use, perceived stress, post-traumatic stress disorder and self-damage was assessed using a χ2 test or *t*-test. Correlations among variables were also calculated. A significance level of <0.05 was used.

The sample constituted a total of 1.062 participants, 77.8% female and 22.2% male. The mean age was 42.45 years old (SD = 13.41), and most participants were between 25–39 years old (33.4%) and 40–54 years old (36.1%).

## 3. Results

Overall, 591 participants (55.6%) reported alcohol consumption in a day during the last four weeks. Most of them (79.2%), reported the consumption of one or two cups, on average, in a day, 15.7% consumed three or four cups, and 5.1% reported the consumption of five or more cups (Table 1).

From the 864 respondents, 33.3% reported the consumption of medication to sleep or calm down. Almost half (42.7%) reported using this medication in the last four weeks, and one third (34.7%) reported the beginning of the consumption only during the first wave of COVID-19 (last four weeks).

Regarding cannabis, 8.4% of the 812 respondents reported its consumption, and 36.8% of them reported the consumption during the last four weeks. Moreover, a quarter (26.5%) of the cannabis users began only in the COVID-19 period. The use of other drugs, such as cocaine, amphetamines, ecstasy and heroin was rare (1.1%), but one third of the users began during the last four weeks (Table 1).

The post-traumatic stress disorder scale was answered by 830 participants (Table 2) with an acceptable internal consistency (α = 0.711). During the last four weeks, 40.8% reported having nightmares about the event(s) or thinking about the event(s) when they did not want to, 46.6% reported trying hard not to think about the event(s) or going out of their way to avoid situations that reminded them of the event(s) and 34.8% answered that they felt numb or detached from people, activities or their surroundings. Moreover, 27.6% declared themselves to be constantly on guard, watchful or easily startled, and 27.4% felt guilty or unable to stop blaming themselves or others for the event(s) or any problems the event(s) may have caused. The average of these items was 1.76. Considering the cut-off value of 3 on the PC-PTSD-5, almost one third (31.5%) of the participants could be considered to have had PTSD. Using a more conservative cut-off value of 4, 18% of the respondents had PTSD (Table 2).

In the Perceived Stress Scale, with a good internal consistency (α = 0.860), respondents obtained a mean of 17.25 (SD = 6.31). Most of the participants were on the moderate level (65.0%). More than a quarter (27.9%) of the respondents had a low level of perceived stress, and 7.1% reached the high levels (Table 3).

The Perceived Stress Scale was highly correlated with the post-traumatic stress disorder scale (r = 0.60, *p* < 0.001).

Suicidal ideation was reported by 17.8% of the participants, suicide was attempted by 8.4%, and self-inflicted damage was reported by 5.1% (Table 4). During COVID-19 there were not reported any suicide attempts. However, suicidal ideation was reported during the last four weeks by 22.3% of all that have reported it, and for 14.9% of them, this kind of ideation began in this period. From the total that reported self-inflicted damage, 10.8% reported during the last four weeks, and 8.1% started this behavior during this period.

Regarding gender differences in substance use, the results show a significant difference between men and women in alcohol (*p* = 0.013) and medication consumption (*p* < 0.001) but not in cannabis (*p* = 0.957). Alcohol consumption during the lifetime was higher in men (63.6%) compared to women (54.5%). On the contrary, concerning medication, women showed a consumption rate of 36.2% compared to 22.9% for men.

The post-traumatic stress disorder level in women was significantly (*p* < *0*.001) higher (M = 1.88, SD = 1.61) than in men (M = 1.33, SS = 1.53). Similarly, the Perceived Stress Scale showed significantly higher values (*p* < 0.001) for women (M = 17.95, SD = 6.19) than for men (M = 14.72, SD = 6.08). Moreover, our results were significantly higher compared to the Portuguese normative values obtained by Trigo et al. [54] for both women (M = 16.6; t(639) = 5.53; *p* < 0.001) and men (M= 13.4; t(176) = 2.90; *p* = 0.004). Lastly, suicidal ideation did not present gender differences (*p* = 0.617).

Concerning the age of the participants, the results show that the mean age of the medication users (M = 44.38, SD = 13.39) was higher compared to those who do not consume medication (*p* = 0.001). On the contrary, cannabis consumers (M = 36.76, SD = 13.04; *p* = 0.001) and individuals with suicidal ideation (M = 39.72, SD = 14.02; *p* = 0.004) tended to be younger. There were no age differences regarding alcohol consumption.

Age was negatively correlated with the Perceived Stress Scale (r = −0.28, *p* < 0.001) and with the post-traumatic stress disorder scale (r = −0.22, *p* < 0.001).

## 4. Discussion

Generally, our study shows a hard pattern of medication use to sleep or calm down since a prevalence rate of 33.3% is higher than the data provided (12.1%) by Portuguese surveys in the general population about substance use [55]. Nonetheless, 42.7% of the medication consumers reported the use of medication during the first wave of COVID-19, and a considerable consumption of medication started only during this period. These rates of medication use were higher in women and older respondents, which is consistent with other studies [56,57,58,59].

Cannabis consumption in the lifetime was slightly less frequent (8.4%) when compared with the data from the same survey (11%) [55]. Almost 37% of the cannabis users consumed in the last four weeks, and more than a quarter started during this period. The cannabis use was not significantly different by gender, but the younger participants reported more. These results are in line with other studies pointing out changes in cannabis consumption during the COVID-19 lockdowns imposed in the first wave, especially concerning the start of cannabis consumption in the general population [60,61,62].

Half of the sample reported alcohol consumption during the last four weeks, which is consistent with the obtained rate for 30 days of consumption (49.1%) [55]. However, only a small percentage (5.1%) of alcohol users present a high pattern of use. The OECD study [61], a multi-country comparison on the effects of COVID-19 on alcohol consumption revealed that, overall, there were not many changes in how much people drank, but particular concern was raised regarding the fact that among those who did, a larger proportion of people drank more. Our research reveals no age differences regarding alcohol consumption, but other studies [62,63] concluded that the alteration of alcohol consumption after the implementation of the lockdown was significantly influenced by age, with mature adults associated with more changes than younger adults.

Our findings demonstrate a rate of 31.5% for post-traumatic stress disorders. These data are higher than the ones found in other meta-analyses [28,29,31,64,65] and were almost always over the maximum value of the CI. The study of Cénat et al. [28] examining several short-term mental health consequences of COVID-19, concluded a 21.94% prevalence of PTSD, while Al Falasi et al. [29] reported a general public prevalence of 19.3%. Moreover, Bourmistrova et al. [27], based on COVID-19 patients, estimated a PTSD prevalence of 17.68%, Nagarajan et al. [31] found a prevalence of 16% of PTSD in COVID-19 survivors and Yunitri et al. [30] estimated of PTSD prevalence of 17.52%.

The Perceived Stress Scale also showed higher values compared to the normative data obtained for men and women in previous Portuguese research [54], which means that the pandemic situations probably enhanced the experienced stress of individuals.

Another problematic issue is the high percentage of the participants (17.8%) reporting suicidal ideation, and some of them maintained or started this kind of ideation during the COVID-19 period. Contrary to the data found in this study with no gender differences and suicidal ideation being more prevalent in young people, official figures on mortality due to suicide in Portugal in pre-pandemic times were growing and affecting mostly male individuals, older people over 75 years old and those living in rural areas [65].

Globally, our results match the scientific evidence about the short-term impact caused by pandemic situations and reinforces the strong implication on mental health of the restrictive conditions created by the public responses to COVID-19. Despite some specificities that were at least partially derived from the sampling procedures, this wider conclusion remains valid. For instance, the medication use reported in this study is considerably higher when compared with the general population prevalence rates, and only a part of this discrepancy is due to a gender bias.

These findings clearly prove the need to care for the mental health of the people, applying suitable health policies before and while subjecting a large proportion of the population to hard stressor events. Nonspecific preventive efforts arise from a better mental health community service. People can be prepared to respond. However, in extreme conditions, some additional efforts should be implemented, such as emotional help lines, alternative access to mental health care services or professional support.

### Limitations of the Study

This study has potential limitations. It is based on an online questionnaire, and due to the sampling procedure, results may not reflect the situation of the population in general. Volunteer bias cannot be excluded since it might have particularly included those with digital literacy. This could explain the lower participation of older groups as well as the under-representation of individuals with lower educational qualifications. Additionally, the sample was also female-dominant, with male under-representation. Thus, all data should be carefully analyzed.

The assessment of mental problems was made exclusively from self-report measures, which can also lead to bias related to the capacity of participants to make accurate self-evaluations and also raise questions about response honesty.

The cross-sectional nature of this study can also challenge inferences about causality. Problems arise from the need to know if identified mental health problems are a consequence of the pandemic situation, derive from previous related difficulties or both. Longitudinal research is deeply encouraged here.

Moreover, the study only focused on short-term effects, and medium- and long-term impact studies would be necessary.

## 5. Conclusions

It has been almost two years’ time since the first COVID-19 wave affected populations worldwide. As our study shows, it has resulted in substantial mental health challenges in Portugal, which had a high prevalence of mental health problems in the general population before the pandemic outbreak. Focused on the self-reported mental health symptoms during the first lockdown, we found changes in medication use to sleep or calm down, cannabis use and perceived stress, which were unevenly distributed across the population. Some impacts could be magnified among vulnerable groups, including women, young adults and older individuals, as previously described. No significant pattern changes were registered in alcohol consumption. Consistent with prior studies in the Portuguese population, suicidal ideation was reported by a high percentage of participants.

These results suggest the need to bridge the pre-existent gap in access to mental health services in Portugal since the pandemic impacts created a greater need for mental health in primary care services. There is a need for further research focused on the long-term mental health impact of COVID-19, which may take months or even years to arise. For a better understanding of these results, future research should include representative samples and also consider treatment strategies adopted by health services to tackle mental health issues, such as the long-term use of medication to calm down and sleep that has started or been aggravated during the lockdown.

## Figures and Tables

**Table 1 ijerph-19-06489-t001:** Substance use during the first wave of COVID-19.

	Total *n* (%)
Alcohol consumption in a day (*n* = 1.062)	
Yes, during the last 4 weeks	591 (55.6)
1 or 2 cups	468 (79.2)
3 or 4 cups	93 (15.7)
5 or more cups	30 (5.1)
Consumption of medication to sleep or calm down (*n* = 864)	
Yes, in lifetime	288 (33.3)
Yes, during the last 4 weeks	123 (42.7)
Yes, only during the last 4 weeks	100 (34.7)
Consumption of cannabis (*n* = 812)	
Yes, in lifetime	68 (8.4)
Yes, during the last 4 weeks	25 (36.8)
Yes, only during the last 4 weeks	18 (26.5)
Consumption of cocaine, amphetamines, ecstasy, heroin or other drugs (*n* = 817)	
Yes, in lifetime	9 (1.1)
Yes, during the last 4 weeks	3 (33.3)
Yes, only during the last 4 weeks	3 (33.3)

**Table 2 ijerph-19-06489-t002:** Post-traumatic stress disorder during the first wave of COVID-19.

	Total *n* (%)
PC-PTSD-5 (*n* = 830; α = 0.711)	Yes
Had nightmares about the event(s) or thought about the event(s) when you did not want to (*n* = 848)	346 (40.8)
Tried hard not to think about the event(s) or went out of your way to avoid situations that reminded you of the event(s) (*n* = 845)	394 (46.6)
Been constantly on guard, watchful, or easily startled (*n* = 841)	232 (27.6)
Felt numb or detached from people, activities, or your surroundings (*n* = 844)	294 (34.8)
Felt guilty or unable to stop blaming yourself or others for the event(s) or any problems the event(s) may have caused (*n* = 843)	231 (27.4)
PC-PTSD-5 Total score (M = SD)	1.76 (1.61)
PC-PTSD-5 Level	
0	251 (30.2)
1	172 (20.7)
2	145 (17.5)
3	112 (13.5)
4	89 (10.7)
5	61 (7.3)

**Table 3 ijerph-19-06489-t003:** Perceived stress during the first wave of COVID-19.

	Total *n* (%)
PSS 10 (*n* = 817; α = 0.860)-(M = SD)	17.25 (6.31)
PSS 10 level	*n* (%)
Low	228 (27.9)
Moderate	531 (65.0)
High	58 (7.1)

**Table 4 ijerph-19-06489-t004:** Suicidal ideation, suicide attempts and self-inflicted damage during the first wave of COVID-19.

	Total *n* (%)
Suicidal ideation (*n* = 830)	
Yes, in lifetime	148 (17.8)
Yes, in last 4 weeks	33 (22.3)
Yes, only in last 4 weeks	22 (14.9)
Suicide attempt (*n* = 804)	
Yes, in lifetime	20 (8.4)
Yes, in last 4 weeks	0
Yes, only in last 4 weeks	0
Self-inflicted damage (*n* = 791)	
Yes, in lifetime	37 (5.1)
Yes, in last 4 weeks	4 (10.8)
Yes, only in last 4 weeks	3 (8.1)

## Data Availability

The data presented in this study are available on request from the last author. The data are not publicly available due to confidentiality reasons.

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
