# Peer review of "The Implication of the First Wave of COVID-19 on Mental Health: Results from a Portuguese Sample"

_ijerph, 2022, doi:10.3390/ijerph19116489_

Round 1

Reviewer 1 Report

This is a great manuscript. I would accept the papers after several minor revisions as follow:

  1. I suggest to add several tables regarding the result of the statistical analysis. Ex: correlation table, statistical analysis table
  2. “The perceived stress scale also shows higher values compared to normative data ob-315 tained for men and women in previous Portuguese research [64].” Try to add more explanations in this paragraph.
  3. What is the theoretical contributions of this study? You may add several paragraphs in the section 4.1
  4. What is the practical implications of this study? You may add several paragraphs in the section 4.2
  5. The limitation of the study is written poorly. The authors need to rewrite this section

Author Response

Dear Reviewer, thank you for your comments.

The answer are attached.

Reviewer 2 Report

This is a very well-written paper and important in the context of understanding the effects of the pandemic on mental health. The literature review in the introduction is state of the art and the importance of the research for the Portugal situation is well described. My major concern is about the data. It is a convenience sample in the context of a wider study. This wider study introduced at the end of the introduction needs more specifications.  (domestic violence?).  Please describe this study in more detail and explain in the methods the pros and cons of convenience sampling for this purpose. Convenience sampling can be very adequate to get a quick impression of research topics and are very helpful to generate ideas to be tested in other (random) sampling frames. But point estimators (such as percentages) could be easily inflated by the chosen method (online convenience sample). In this case, voluntarism is probably not about illiteracy but more in the direction of recruiting interested participants (who have mild or moderate complaints) and because of this are more interested in participating. So the sampling frame is in this case a potential problem for interpreting the found results. My suggestion would be to strengthen the paper by discussing the found figures in relation to the sampling procedures. A good discussion of this topic is very fruitful for the readers because many scientist are struggling with this: there is pressure on science to give a quick idea of what is happening in society during the pandemic and this means some compromises.

Minor

Check figures 3,2 or 3.2 (for example line 177: 4,4)

Please do not use P<0.01, but give exact p values for all tests for meta-analytic  purposes

Author Response

Dear Reviewer,

Thank you for your comments. The answers are attached.

Reviewer 3 Report

This is an interesting study albeit only on a subset of Portugese individuals. The paper needs editing for spelling, use of articles and English expressions some examples

Line 18 insert “the” COVID-19

Line 19 Remove “the” individuals

Line 19 Replace with Meta-analysis indicate

Also it is not shared why Portugal has such high rates of mental health problems and Lines 300-302 are confusing in terms of summarizing some of the findings.

Author Response

(The authors gave the same response as above.)
